

# Dissemination of antibiotic resistance genes associated with the sporobiota in sediments impacted by wastewater

Christophe Paul[1,*], Zhanna Bayrychenko[1,*], Thomas Junier[2], Sevasti Filippidou[1], Karin Beck[3], Matthieu Bueche[1], Gilbert Greub[4], Helmut Bürgmann[3] and Pilar Junier[1]

[1] Institute of Biology, Laboratory of Microbiology, University of Neuchatel, Neuchâtel, NE, Switzerland
[2] Vital-IT, Swiss Institute of Bioinformatics, Lausanne, Switzerland
[3] Eawag, Swiss Federal Institute of Aquatic Science and Technology, Kastanienbaum, Switzerland
[4] Institute of Microbiology, University Hospital Center, University of Lausanne, Lausanne, Switzerland
[*] These authors contributed equally to this work.

Corresponding author
Pilar Junier, pilar.junier@unine.ch

## ABSTRACT

Aquatic ecosystems serve as a dissemination pathway and a reservoir of both antibiotic resistant bacteria (ARB) and antibiotic resistance genes (ARG). In this study, we investigate the role of the bacterial sporobiota to act as a vector for ARG dispersal in aquatic ecosystems. The sporobiota was operationally defined as the resilient fraction of the bacterial community withstanding a harsh extraction treatment eliminating the easily lysed fraction of the total bacterial community. The sporobiota has been identified as a critical component of the human microbiome, and therefore potentially a key element in the dissemination of ARG in human-impacted environments. A region of Lake Geneva in which the accumulation of ARG in the sediments has been previously linked to the deposition of treated wastewater was selected to investigate the dissemination of *tet*(W) and *sul*1, two genes conferring resistance to tetracycline and sulfonamide, respectively. Analysis of the abundance of these ARG within the sporobiome (collection of genes of the sporobiota) and correlation with community composition and environmental parameters demonstrated that ARG can spread across the environment with the sporobiota being the dispersal vector. A highly abundant OTU affiliated with the genus *Clostridium* was identified as a potential specific vector for the dissemination of *tet*(W), due to a strong correlation with *tet*(W) frequency (ARG copy numbers/ng DNA). The high dispersal rate, long-term survival, and potential reactivation of the sporobiota constitute a serious concern in terms of dissemination and persistence of ARG in the environment.

## INTRODUCTION

The prevalence and spread of antibiotic resistance is a pressing global public health issue (*Marti, Variatza & Balcazar, 2014*; *Perry, Westman & Wright, 2014*; *O'Neill, 2015*). Although resistance to antibiotics was traditionally viewed as a clinical problem, attention

has recently been directed at understanding the ecological and environmental processes involved in the dissemination of antibiotic-resistant bacteria (ARB) and their associated resistance genes (ARG) (*Berglund, 2015*; *Bengtsson-Palme, Kristiansson & Larsson, 2017*). The dispersal potential of organisms (or genetic elements) carrying ARG is an important element to understand dissemination of antibiotic resistance in the environment. In terms of dissemination, the so-called "sporobiota" is a portion of the bacterial community of great interest. The sporobiota has been defined as a fraction of the microbial community that can exist in the form of highly transmissible spores, which are spread in the environment and are implicated in host-to-host transmission (*Tetz & Tetz, 2017*). This fraction of the community, which can be considered as part of the seed bank (*Lennon & Jones, 2011*; *Shoemaker & Lennon, 2018*), is prone to high dispersal rates due its non-active physiological state, bypassing limitations for local adaptation (*Bartholomew & Paik, 1966*; *Hubert, 2009*; *Lennon & Jones, 2011*). The sporobiota appears to play a significant role as part of the natural microbiota of humans (*Browne et al., 2016*), and therefore, might be highly relevant for ARG dissemination in human-impacted environments. Moreover, one of the features that has been ascribed to the sporobiome (collection of genes of the sporobiota community) is its potential implication in the spread of antibiotic resistance (*Tetz & Tetz, 2017*; *Bengtsson-Palme, Kristiansson & Larsson, 2017*). Although the initial definition of the sporobiota in humans makes reference to a particular type of spores produced by the phylum Firmicutes (heat-resistant endospores; *Tetz & Tetz, 2017*), it is likely that other spore-like structures are found as part of the environmental sporobiota. Therefore, in this study we use a more comprehensive operational definition based on the properties of the sporobiota. The operational definition used here consists of cellular structures withstanding a harsh extraction method designed originally to enrich in endospores from environmental samples (*Wunderlin et al., 2014*).

Aquatic ecosystems can be considered as priority areas when investigating dispersal of ARG in the environment. On the one hand, aquatic ecosystems have a reservoir function, allowing the mixing of environmental organisms and human/animal pathogens, potentially promoting gene transfer, mainly in sediments and biofilms (*Van Elsas & Bailey, 2002*; *Kenzaka, Tani & Nasu, 2010*; *Drudge, 2012*). On the other hand, the use of water bodies for irrigation, recreational and domestic purposes, and as a drinking water supply (after treatment), constitutes a potential pathway enabling transmission of ARB (or ARG) between hosts through the environment (*Baquero, Martínez & Cantón, 2008*; *Taylor, Verner-Jeffreys & Baker-Austin, 2011*). In particular, water bodies connected to wastewater treatment plants (WWTPs) are a major concern (see *Baquero, Martínez & Cantón, 2008*; *Marti, Variatza & Balcazar, 2014* for reviews in the topic). Many WWTPs not only collect wastewater from domestic sources, but also from hospitals or the food-industry, where antibiotics are extensively used (*Szczepanowski et al., 2009*; *Bouki, Venieri & Diamadopoulos, 2013*; *Rodriguez-Mozaz et al., 2015*; *Xu et al., 2015*). Despite elaborate treatment and disinfection phases put in place in modern WWTPs, ARB and ARG have been shown to be only partially eliminated, thus potentially having an important impact on ARG transfer and dispersal (*LaPara et al., 2011*; *Munir, Wong & Xagoraraki, 2011*; *Dodd, 2012*; *Rizzo et al., 2013*).

Lake Geneva is the largest freshwater lake in Western Europe and constitutes a major source of drinking water. The lake receives wastewater from the surrounding cities. The largest WWTP is located near the city of Lausanne and receives wastewater from both domestic and industrial/clinical sources. Treated wastewater is released into a bay of the lake, the Vidy Bay. Overflow associated with strong rain episodes also contributed to the occasional release of untreated wastewater. Although a natural background of resistance genes is ubiquitous, and in addition the entire Lake Geneva basin is likely affected by human activity, sediments of the Vidy Bay are more significantly impacted in terms of organic and inorganic pollution by the WWTP discharge, due to their immediate proximity to the outlet. Contamination by fecal-indicator bacteria, trace metals, nitrogen, phosphorus, and ARG has been reported (*Haller et al., 2011*; *Czekalski et al., 2012*; *Thevenon et al., 2012*; *Czekalski, Gascón Díez & Bürgmann, 2014*; *Devarajan et al., 2015*). Moreover, ARG and resistant bacteria have been detected close to the drinking water pump of Lausanne, 3.2 km away from the WWTP outflow, highlighting the potential risk for human health from the transfer of ARB and ARG from the environment back into humans (*Czekalski et al., 2012*). Therefore, Lake Geneva constitutes an ideal system to evaluate the role of the sporobiota in ARG dissemination. We have previously shown that the sporobiome of lake sediment preserves a historical record of resistance prevalence using as proxies for ARG the *tet*(W) and *sul*1 genes (*Madueño et al., 2018*) and the same genes were also selected for this study. Although analyzing only two ARG limits the scope of our work and results should therefore not be generalized, currently the low DNA yield resulting from the separation of the sporobiota limits the number of genes that can be assessed simultaneously without total DNA amplification. The *tet*(W) gene confers resistance to tetracycline, a class of natural antibiotics isolated from *Streptomyces* that inhibit protein synthesis (*Roberts, 1996*). The *sul*1 gene confers resistance to sulfonamide, a class of synthetic antibiotics that inhibit the enzyme dihydropteroate synthase (DHPS) in the folic acid pathway (*Sköld, 2001*). Moreover, the *sul*1 gene has been suggested as a useful proxy for monitoring ARG in the environment (*Berendonk et al., 2015*). Both genes are known to be abundant in wastewater-impacted sediments (*Czekalski, Gascón Díez & Bürgmann, 2014*; *Na et al., 2014*; *Rodriguez-Mozaz et al., 2015*), and in particular, the high abundance of these two genes among total bacterial communities was previously reported in sediments from the same location (*Czekalski, Gascón Díez & Bürgmann, 2014*). Sulfonamides and tetracyclines are still in use in human medicine and are the antibiotic classes with the highest and third highest sales numbers, respectively, for the veterinary sector in Switzerland (*Federal Office of Public Health, 2016*).

The aims of the present study were to investigate the accumulation of ARG *tet*(W) and *sul*1 in DNA extracted from sediments treated to enrich the sporobiota and that are impacted differentially by a WWTP discharge. In addition, we analyzed the sporobiota community composition and its relationship to ARG levels, spatial distribution, and characteristics of sediments.

## MATERIAL AND METHODS

### Site description and sampling

Lake Geneva is the largest freshwater lake of Western Europe, with a volume of 89 $km^3$, a surface area of 580 $km^2$ and a maximum depth of 309 m. Vidy Bay is enclosed by the shoreline near Lausanne between St.-Sulpice and Lausanne-Ouchy. The outlet pipe of the WWTP is located 700 m offshore at 35 m depth (46.51197121 N; 6.587423025 E).

The ten cores used for this study were retrieved using standard corers (50 cm length Plexiglas tubes of 6 cm diameter) between July 2011 and May 2012 in three sampling zones differently impacted by the WWTP (*Bueche, 2014*; *Sauvain et al., 2014*): 5–40 m from the outlet pipe of the WWTP ("near", N1–N4); 134–429 m from the outlet pipe ("middle", M1–M4); and 611–956 m from the outlet pipe ("distal", D1–D2). Two sediment layers were analyzed for the cores with the code numbers 1 and 2 (D1, D2, M1, M2, N1, N2) [20]: 0–3 cm and 3–9 cm depth. Three sediment layers were analyzed for the cores with the code numbers 3–4 (M3, M4, N3, N4): 0–1.5 cm, 1.5–3 cm and 3–9 cm depth.

### DNA extraction

Sporobiome DNA was obtained using an indirect three step extraction method: (i) extraction of cells from the sediment particles, (ii) separation of spores from the vegetative cells, (iii) DNA extraction from the spores (*Wunderlin et al., 2014*).

The treatment to separate spores from vegetative cells was performed on the biomass from 1.5 grams of sediment collected on one nitrocellulose filter (Merck Millipore, Darmstadt, Germany) per sample, as previously described (*Wunderlin et al., 2014*). The first step consisted of the lysis of vegetative cells by heat, enzymatic agents (lysozyme) and chemicals (Tris-EDTA, NaOH, SDS). The second step consisted of a DNase digestion in order to destroy the free DNA. Treated filters were stored at −20 °C until DNA was extracted from the pre-treated filters using a protocol based on the FastDNA® SPIN kit for soil (MP Biomedicals, Santa Ana, CA, USA) (*Wunderlin et al., 2013*) with the following additional modifications: samples in lysing matrix were submitted to two successive bead-beating steps. Supernatant retrieved from each bead-beating was treated separately according to manufacturer's instructions and DNA extracts were pooled together at the end of the procedure. Pooled DNA was precipitated with 0.3 M Na-acetate and ethanol (99%), stored at −20 °C overnight and centrifuged for 1 h at 21460xg and 4 °C. Supernatant was removed and the pellet was washed with 1 ml of 70% ethanol and centrifuged for 30 min at 21460xg and 4 °C. Supernatant was removed and the residual ethanol was allowed to evaporate at room temperature. Pellet was re-suspended in 50 µl of PCR-grade water. Total DNA was quantified using Qubit® dsDNA HS Assay Kit on a Qubit® 2.0 Fluorometer (Invitrogen, Carlsbad, CA, USA).

### Real-time quantitative PCR on *tet(W)* and *sul*1 genes

Real-time Taqman®-PCR on *sul*1 and *tet*(W) genes was performed in 384-well plates using a LightCycler®480 Instrument II (Roche, Basel, Switzerland). For *sul*1, the primers used were qSUL653f (5′-CCGTTGGCCTTCCTGTAAAG-3′) and qSUL719r

(5′-TTGCCGATCGCGTGAAGT-3′) with tpSUL1 (*FAM*-CAGCGAGCCTTGCGGCGG-TAMRA) probe (*Heuer & Smalla, 2007*). The reaction mix for *sul*1 consisted of 2 μL of DNA template (between 0.08 and 1.39 ng/μL), 0.025 μM of each primer, 0.25 μM of TaqMan probe and 1 × TaqMan® Fast Universal PCR Master Mix (Applied Biosystems, Foster City, CA, USA). Total reaction volume of 10 μL was reached with PCR-grade water. For *tet* (W), the primers used were tetW_f (5′-CGGCAGCGCAAAGAGAAC-3′) and tetW_r (5-CGGGTCAGTATCCGCAAGTT-3′) with tetW_s (*FAM*-CTGGACGCTCTTACG-TAMRA) probe (*Walsh et al., 2011*). The reaction mix for *tet* (W) consisted of 2 μL of DNA template, 0.025 μM of each primer, 0.1 μM of TaqMan probe and 1 × TaqMan® Fast Universal PCR Master Mix (Applied Biosystems, Foster City, CA, USA). Total reaction volume of 10 μL was reached with PCR-grade water. The qPCR program was the same for both genes and started with a hold at 95 °C for 10 min, followed by 45 cycles of denaturation at 95 °C for 15 s and annealing/elongation at 60 °C for 1 min. The qPCR assays were performed in technical triplicates on samples, standards and negative controls. The negative controls consisted of PCR blanks with only the reaction mix and of PCR blanks containing the mix and 2 μL of PCR-grade water. Standard curves were prepared from serial 10-fold dilutions of plasmid DNA containing the respective target gene in a range of $5 \times 10^7$ to 50 gene copies. For *sul*1, control plasmids and standard curves were prepared as previously described (*Heuer & Smalla, 2007*). For *tet* (W), standard curves were prepared as previously described (*Walsh et al., 2011*). The effect of inhibitors on amplification was tested for all the samples and for both genes. All samples were spiked with $10^4$ copies of plasmid DNA containing the *tet* (W) or the *sul*1 gene and amplified together with the same set of non-spiked samples and control DNA and the results indicated that inhibition was negligible.

## Sequencing and data analysis

Purified DNA extracts were sent to Fasteris (Geneva, Switzerland) for 16S rRNA gene amplicons sequencing using Illumina MiSeq platform (Illumina, San Diego, USA), generating 250 bp paired-end reads. The hypervariable V3–V4 region was targeted using universal primers Bakt_341F (5′-CCTACGGGNGGCWGCAG-3′) and Bakt_805R (5′-GACTACHVGGGTATCTAATCC-3′) (*Herlemann et al., 2011*). Analysis of the dataset was performed using Mothur (*Schloss et al., 2009*) following the MiSeq SOP (*Kozich et al., 2013*). The SILVA NR v123 reference database (*Quast et al., 2013*) was used for the alignment of amplicons and the taxonomic assignment of representative OTUs. After quality filtering and removal of chimeras, a total of 1818238 amplicons were obtained (408758 unique sequences). Singletons were removed prior to the clustering into OTUs and corresponded to 373374 sequences. Average neighbor clustering of the 1444864 remaining sequences (35384 unique sequences) with an identity threshold of 97% led to the identification of 6390 OTUs. Sequencing data was deposited to NCBI under the Bioproject accession number PRJNA396277.

## Statistical and multivariate analysis

Community and statistical analyses were performed using R version 3.4.0 (*R Core Team, 2014*) and the *phyloseq* and *vegan* packages (*McMurdie & Holmes, 2013*; *Oksanen et al.,*

*2017*). The distance of samples to the WWTP outflow is a combination (euclidian distance) of the distance between the sample and the pipe on the east axis, the north axis (based on CH coordinates) and the water column depth. It is expressed in meters. The difference in ARG levels between the zones (near, middle and distal) was tested using analysis of variance (ANOVA) and Tukey's test for pairwise comparisons. Pairwise correlations between environmental parameters (organic carbon, total nitrogen, and concentration of cadmium, copper, iron, manganese, aluminum, zinc and arsenic) and ARG abundance/frequency were calculated using Spearman's rank correlation coefficient. This correlation coefficient was selected because we did not assume a normal distribution of the variables. The organic carbon, total nitrogen, and metal concentrations were obtained from previous studies (*Bueche, 2014*; *Sauvain et al., 2014*). *P*-values were adjusted for family-wise error rate using the Holm method. Spearman's rank correlation coefficient was used for calculating the correlations between ARG frequency and the relative abundance of OTUs. To provide a visual representation of the distribution of the correlation coefficient of OTUs with ARG concentration, Kernel density curves were computed from Spearman's correlation coefficients calculated between each ARG and OTU and can be thought of as continuous histograms representing the frequency (density) of the correlation coefficients. The significance of the Spearman's correlation coefficients was tested by correcting the *p*-values using the false discovery rate correction (Benjamini–Hochberg method). This method was selected because of the large number of comparisons performed. Principal component analysis (PCA) was computed on the environmental parameters and ARG abundance/frequency, after standardization (zero mean and unit variance). The environmental parameters included were $C_{org}$, $N_{tot}$, metals (listed above), distance to the outlet pipe (Dist_pipe), DNA abundance in ng per gram of sediment (DNA_ng_gSed), and ARG abundance and frequency. Sporobiota community was analyzed by principal coordinates analysis (PCoA), based on Bray-Curtis dissimilarity and Hellinger transformation of the OTU table. Environmental parameters and ARG abundance/frequency were standardized and passively fitted to the ordination. Only significant parameters were displayed ($p < 0.05$).

## RESULTS

### Prevalence and spatial distribution of *tet*(W) and *sul*1 in sporobiome DNA

To determine the prevalence of the two selected ARG in DNA extracted from the sporobiota and to describe their spatial distribution in sediments impacted by a WWTP, ARG quantification was carried out for three zones located at 5-40 m (near) 134-429 m (middle) and 611–956 m (distal) from the outlet pipe (Fig. 1). These zones were previously shown to be differentially impacted by the release of wastewater, according to the concentration of particulate trace and heavy metals in the sediments and to the bacterial community composition (*Sauvain et al., 2014*). The results of ARG quantification were expressed in two different units. The first unit, ARG abundance, refers to the number of copies of ARG per g of sediment. This reflects accumulation in sediment and represents the total ARG

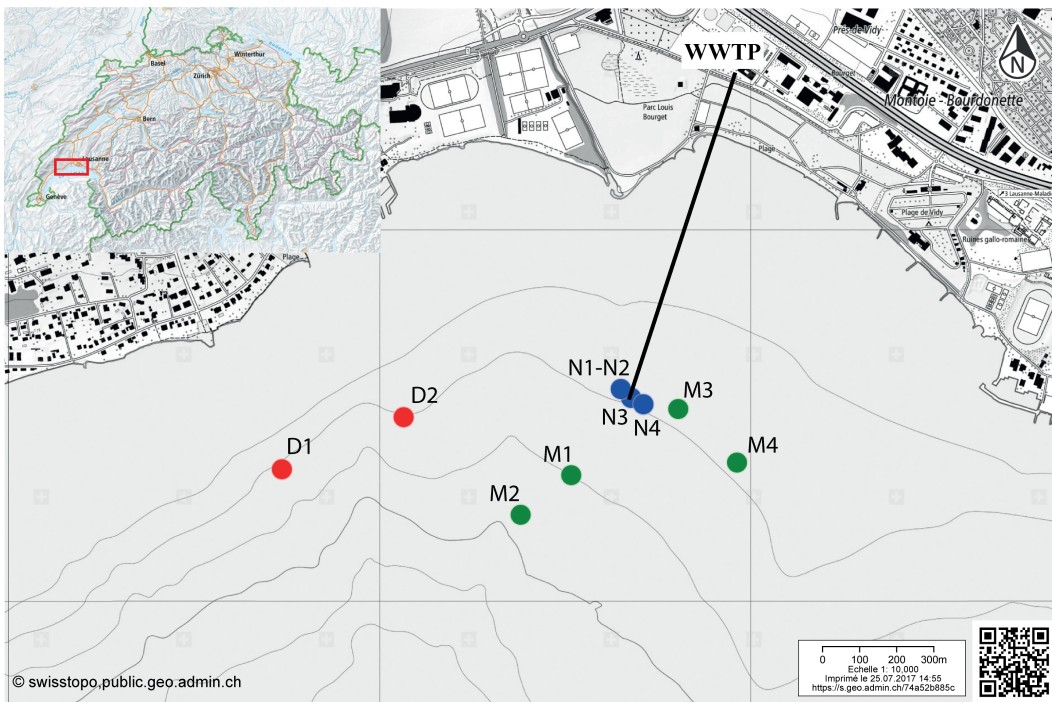

**Figure 1** **Map indicating the sampling locations in the Vidy Bay, Lake Geneva, Switzerland.** Samples were retrieved from three zones differently impacted by the wastewater treatment plant (WWTP). Red circles are for the distal zone (D; 611–956 m from the outlet pipe), green circles are for the middle zone (M; 134–429 m), and blue circle are for the near zone (N; 5–40 m). Outlet pipe of the WWTP is represented by the black line. The original maps were obtained from http://www.swisstopo.ch, owners of the copyright of the image.

pool for a given ARG. However, differential microbial load in sediments could bias this measure. Therefore, we used a second unit, ARG frequency, which is defined as the number of copies of ARG per ng of extracted DNA. This allowed us to estimate the enrichment of specific ARG in DNA from the enriched sporobiota. We caution that both approaches are potentially affected by changes in the proportion of different bacterial groups that may not be causally related to either inputs or growth of ARB or local selection of ARG. For example, an increase in abundance of a population without an ARG due to nutrient inputs will result in a reduction of ARG frequency.

Both ARG, *tet*(W) and *sul*1, were detected in DNA at all sampling locations. In all samples, abundance and frequency were higher for *tet*(W) than for *sul* 1 (Table 1). The influence of the WWTP as the main source of sporobiota-associated ARG can be observed in the spatial distribution of ARG abundance. Abundance was highest in samples from the proximal zone and decreased with distance to the outlet pipe (Fig. 2, Figs. S1 and S2). This pattern was confirmed by principal component analysis (PCA; Fig. S1) and Spearman correlation tests (Fig. S3), which showed that ARG abundance was mostly correlated with environmental variables associated with deposition of treated wastewater, such as organic carbon content ($C_{\text{org}}$) and total nitrogen ($N_{\text{tot}}$) in sediments. However, unexpectedly, the

**Table 1  Abundance and frequency of antibiotic resistance genes in sediments.** Abundance (copies/g of sediment) and frequency (copies/ng of DNA) of two antibiotic resistance genes (*tet*(W) and *sul*1) in the sporobiota, in sediments samples from the Vidy Bay (Lake Geneva, CH).

| Sample | Sediment depth | Dist. to the pipe | DNA | *tet*(W) | | *sul*1 | |
|---|---|---|---|---|---|---|---|
| | (cm) | (m) | (ng/g sed) | (copies/ng DNA) | (copies/g sed) | (copies/ng DNA) | (copies/g sed) |
| D1_low | 3–9 | 955.89 | 3.081 | 3.01E+03 | 9.26E+03 | 4.65E+01 | 1.43E+02 |
| D1_up | 0–3 | 955.89 | 5.385 | 2.81E+03 | 1.51E+04 | 9.33E+01 | 5.02E+02 |
| D2_low | 3–9 | 610.99 | 2.461 | 3.73E+03 | 9.19E+03 | 5.17E+01 | 1.27E+02 |
| D2_up | 0–3 | 610.99 | 3.805 | 3.27E+03 | 1.24E+04 | 8.58E+01 | 3.26E+02 |
| M1_low | 3–9 | 259.15 | 3.066 | 2.56E+04 | 7.85E+04 | 2.29E+02 | 7.02E+02 |
| M1_up | 0–3 | 259.15 | 3.530 | 1.19E+04 | 4.19E+04 | 3.22E+02 | 1.14E+03 |
| M2_low | 3–9 | 429.04 | 2.391 | 2.50E+04 | 5.98E+04 | 1.78E+02 | 4.25E+02 |
| M2_up | 0–3 | 429.04 | 2.166 | 1.19E+04 | 2.58E+04 | 3.82E+02 | 8.26E+02 |
| M3_low | 3–9 | 133.76 | 0.773 | 7.37E+03 | 5.69E+03 | 2.80E+01 | 2.17E+01 |
| M3_med | 1.5–3 | 133.76 | 1.104 | 9.56E+03 | 1.06E+04 | 1.75E+02 | 1.93E+02 |
| M3_up | 0–1.5 | 133.76 | 2.963 | 1.38E+04 | 4.10E+04 | 4.97E+02 | 1.47E+03 |
| M4_low | 3–9 | 335.80 | 0.968 | 8.85E+03 | 8.57E+03 | 1.65E+02 | 1.60E+02 |
| M4_med | 1.5–3 | 335.80 | 1.958 | 2.43E+03 | 4.75E+03 | 6.56E+01 | 1.28E+02 |
| M4_up | 0–1.5 | 335.80 | 1.208 | 8.93E+03 | 1.08E+04 | 5.83E+02 | 7.04E+02 |
| N1_low | 3–9 | 5.39 | 2.313 | 7.24E+03 | 1.67E+04 | 1.26E+02 | 2.92E+02 |
| N1_up | 0–3 | 5.39 | 4.802 | 1.21E+04 | 5.79E+04 | 3.21E+02 | 1.54E+03 |
| N2_low | 3–9 | 5.39 | 4.183 | 1.46E+04 | 6.12E+04 | 2.32E+02 | 9.69E+02 |
| N2_up | 0–3 | 5.39 | 7.658 | 1.81E+04 | 1.38E+05 | 6.20E+02 | 4.75E+03 |
| N3_low | 3–9 | 36.14 | 2.497 | 1.31E+04 | 3.27E+04 | 1.57E+02 | 3.92E+02 |
| N3_med | 1.5–3 | 36.14 | 2.428 | 1.36E+04 | 3.31E+04 | 4.08E+02 | 9.90E+02 |
| N3_up | 0–1.5 | 36.14 | 2.617 | 1.29E+04 | 3.38E+04 | 4.18E+02 | 1.09E+03 |
| N4_low | 3–9 | 39.61 | 13.900 | 1.14E+04 | 1.58E+05 | 7.80E+02 | 1.08E+04 |
| N4_med | 1.5–3 | 39.61 | 10.515 | 1.72E+04 | 1.81E+05 | 1.34E+03 | 1.41E+04 |
| N4_up | 0–1.5 | 39.61 | 7.115 | 1.92E+04 | 1.37E+05 | 1.60E+03 | 1.14E+04 |

$R^2$ calculated for the linear regression established between ARG abundance and distance to the outlet pipe in our case (Fig. S2) was generally low.

Measures of ARG frequency showed that *tet*(W) and *sul*1 genes had different enrichment patterns in the sporobiota. In the case of *sul*1, ARG frequency decreases in relation to the distance to the outlet pipe (Fig. 2, Fig. S2). A decrease in frequency could be attributed to dilution of a population with high *sul*1 frequency (originating from the wastewater) within the environmental sporobiota community. The frequency of *tet*(W) also decreased in relation to the distance to the outlet pipe, but its pattern of distribution was more complex. The highest *tet*(W) frequencies were found in two middle zone samples (M1_low and M2_low, Fig. 2, Fig. S1). Both correlation tests (Fig. S3) and PCA with the environmental variables (Fig. S1B) indicated a significant correlation between *tet*(W) frequency and certain metals (Fe, As, and Cu).

## Sporobiota community composition

Several studies have demonstrated a strong impact of WWTP discharge on bacterial community composition, which is shown by changes in the community in impacted

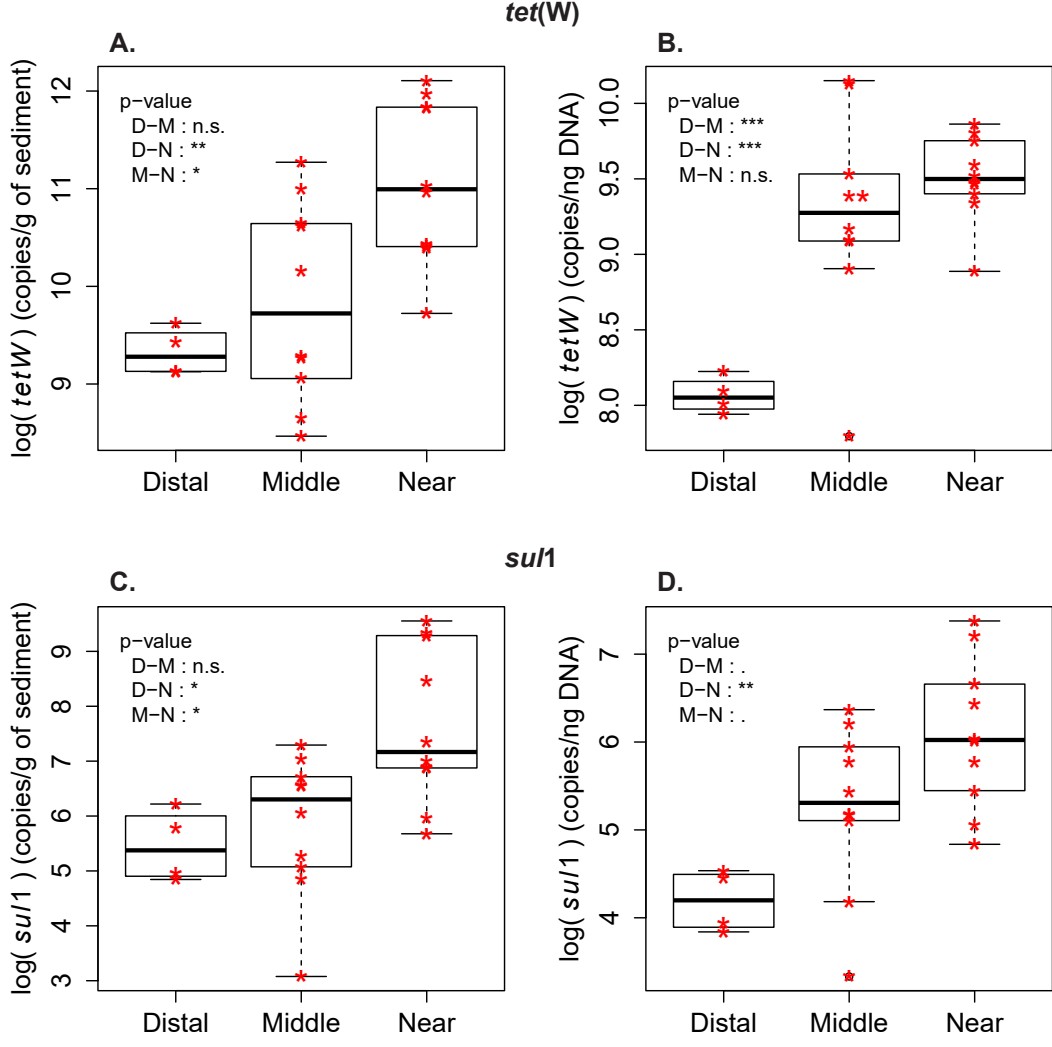

**Figure 2  Abundance and frequency of ARG in relationship to distance to the wastewater treatment plant.** Boxplots for *tet*(W) and *sul*1 distribution in the samples from distal [D], middle (M) and near (N) sampling zones. (A) *tet*(W) log-transformed gene abundance (copies/g of sediment). (B) *tet*(W) log-transformed gene frequency (copies/ng DNA). (C) *sul*1 log-transformed gene abundance (copies/g of sediment). (D) *sul*1 log-transformed gene frequency (copies/ng DNA). Pairwise comparison with Tuckey's test was performed to compare the ARG level between the zones (Distal-Middle, Distal-Near, Middle-Near). Significance codes of *p*-values: $0 < *** < 0.001 < ** < 0.01 < * < 0.05 < . < 0.1$. (n.s.) stands for "not significant".

sediments compared to remote sites (*LaPara et al., 2011*; *Czekalski, Gascón Díez & Bürgmann, 2014*; *Sauvain et al., 2014*). In previous studies using a different sequencing technology (pyrosequencing), the total bacterial communities analyzed from the same sediments were highly divergent, forming at least four distinct groups depending on the distance to the outlet pipe and the concentration of heavy and trace metals in the sediments (*Bueche, 2014*; *Sauvain et al., 2014*). In contrast, the sporobiota community has a low degree of variability among impacted and supposedly less impacted sites (Fig. 3), suggesting that
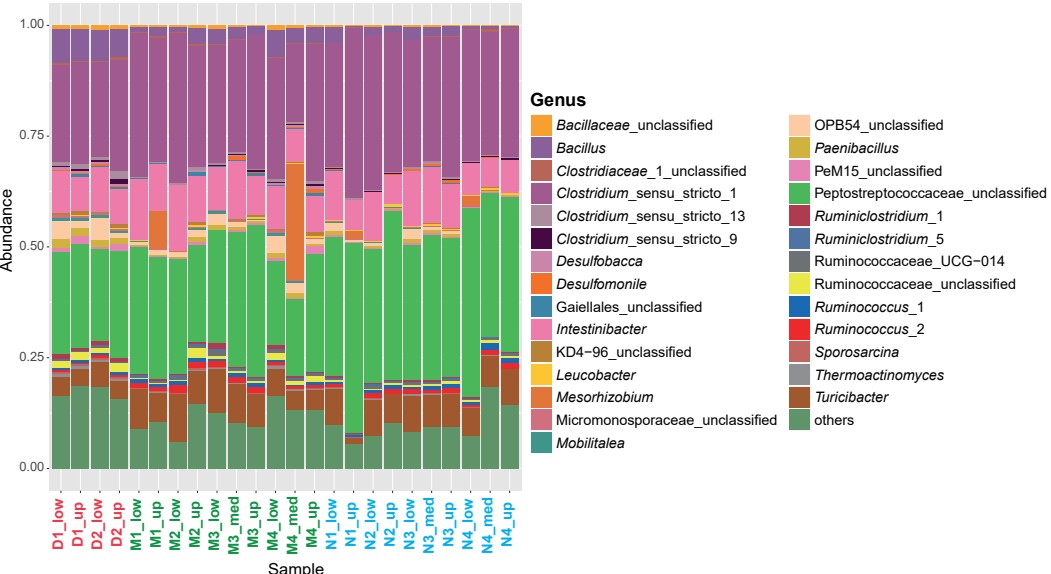

**Figure 3** **Characterization of the sporobiome community in sediments.** Composition of the sporobiome community in sediments from Vidy Bay (Lake Geneva) based on 16S rRNA gene amplicon sequencing. Relative abundance of the most abundant genera (>0.2% of the total community) is represented.

the distance to the wastewater outlet pipe had a relatively small impact on sporobiota community composition. Firmicutes are the dominant phylum in all samples (Fig. S4), accounting for at least 85.2% of the sporobiota community, except in sample taken from the middle area at a depth of 1.5–3 cm (M4_med; 65.9%). This is in agreement with a previous study demonstrating an enrichment of 83.9–90.6% in Firmicutes using the same method for the isolation of spores (*Wunderlin et al., 2014*). Among the non-Firmicutes representatives, we found several groups for which the production of spores is so far not reported and that could represent potential contamination with vegetative cells that withstood the extraction procedure (see below). Nevertheless, we performed the analysis with the entire community as our definition of the sporobiota considers the resilience to the extraction method, rather than phylogenetic affiliation.

In terms of relative abundance, 43.1% of the whole community was identical between all samples. The two most abundant genera, *Clostridium*_sensu_stricto_1 and an unclassified Peptococcaceae, were the same for all samples (except M4_med), with an average relative abundance of 57.3%. Both are widely represented in the human gut (*Lozupone et al., 2012*), which is also the case for other subdominant genera, including *Intestinibacter* (9.7%), *Turicibacter* (6.7%), and *Ruminococcus*_2 (0.9%). These findings suggest a common origin of the sporobiome and point again at the WWTP as the main source of the sporobiome community depositing in the lake sediment in the area studied here.

Despite the high homogeneity of the sporobiota community, principal coordinate analysis (PCoA, Fig. 4) reveals a slight variability among the communities. For example, in Axis 1, Bacilli is associated with samples located at a greater distance from the outlet pipe,

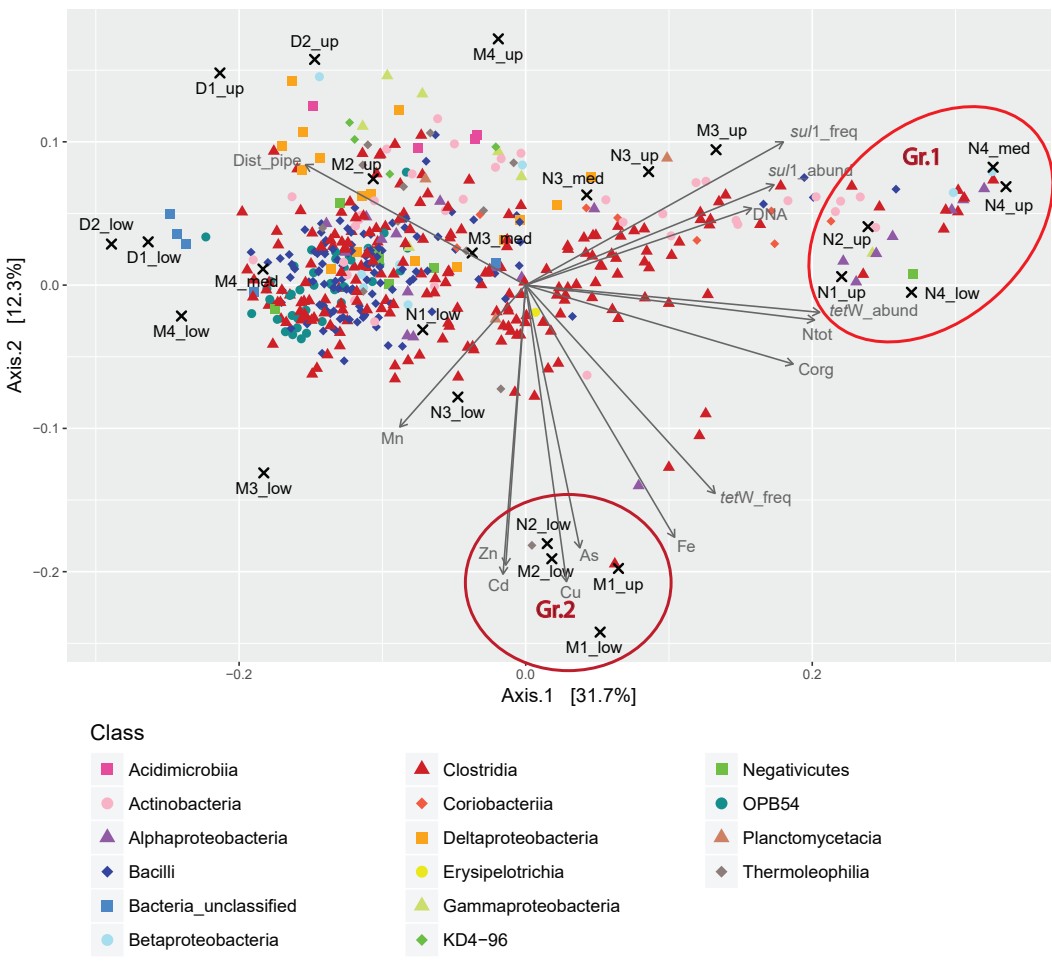

**Figure 4  Principal coordinates analysis (PCoA) triplot of the sediment samples based on Bray-Curtis dissimilarity and Hellinger transformation of the OTU table.** OTUs represented by less than four sequences in the whole dataset were removed from the analysis. OTUs were classified at genus level (or higher taxonomic rank if not possible). Colors correspond to different orders. Only the 80 most abundant OTUs are shown. Environmental variables (including $C_{org}$, $N_{tot}$, trace and heavy metals (TMs), the distance to the outlet pipe (Dist_pipe), DNA abundance in ng/g sediment (DNA_ng_gSed), and ARG abundance (cp_gSed) and frequency (cp_ng)) were standardized and passively fitted to the ordination. Only significant parameters were displayed ($p < 0.05$). Values for $C_{org}$, $N_{tot}$, and TMs were obtained from previous studies (*Bueche, 2014*; *Sauvain et al., 2014*) .

while different Clostridia are associated to samples closer to the WWTP. While Clostridia are mostly anaerobic and are common in the microbiome of humans and other mammals, Bacilli include numerous aerobic organisms and are commonly found in soils (*Madigan et al., 2015*). Accordingly, Clostridia could be seen as a signature of the human influence in the sporobiota associated with wastewater discharge whereas Bacilli would represent a more environmental sporobiota community.

In addition, PCoA revealed two groups of samples of particular interest (Fig. 4). Group 1 (Gr.1 in Fig. 4; Fig. S5B), which was characterized by high load of $C_{org}$ and $N_{tot}$, and high abundance/frequency of ARG, was composed of samples from the zone in close

proximity to the WWTP discharge. Among the OTUs associated to this group, the most abundant belong to the genera *Bifidobacteria* and *Collinsella*, two Actinobacteria common in the human microbiome (*Lamendella et al., 2008*; *Thorasin, Hoyles & McCartney, 2015*), albeit unknown for the production of spores. Another abundant genus in this group is *Trichococcus*, a Firmicute found in sewerage infrastructure and WWTP (*Vandewalle et al., 2012*). This group reflects the direct influence of the WWTP (Fig. S5B). In contrast, Group 2 (Gr.2 in Fig. 4; Fig. S5C) clustered samples from both the near and middle zones and was related to elevated concentrations of particulate metals (Zn, Cd, Cu, As, and Fe) and a high frequency of *tet*(W) (Fig. 4). OTUs defining this group belonged mainly to non-Firmicutes (Fig. S5C). Some were affiliated to organisms that have been previously isolated from WWTPs, including Synergistetes (*Wang et al., 2013*), *Syntrophorhabdus* (*Saia et al., 2016*), Anaerolinaceae and *Leptolinae* (*Yamada et al., 2006*). Others were affiliated to uncommon taxa previously isolated from rivers or aquifers, such as Gaiellales (*Albuquerque et al., 2011*) and *Kaistia* (*Jin et al., 2011*). To our knowledge, there is no evidence that these organisms are resistant to high metal contamination.

## Correlation tests between OTUs and ARG

To further investigate the relationship between the sporobiota community and ARG distribution, pairwise spearman correlation tests were performed between *tet*(W) frequency and the relative abundance of each OTU. For both *tet*(W) and *sul*1, the results indicated that only a small number of OTUs correlate with the ARG studied here (Figs. 5A–5B). Interestingly, the 10 most positively correlated OTUs were specific to each ARG. Although environmental correlations alone are not sufficient to identify carriers of antibiotic resistance, the results can help to highlight potential groups that merit further investigation. For example, the three most *tet*(W)- positively correlated OTUs (correlation coefficients ranging from 0.84 to 0.71) all belonged to Firmicutes (Table 2). In particular, one of these OTUs (OTU00002; *Clostridium*_sensu_stricto_1) was highly abundant (over 20% of the relative abundance) in all samples (Fig. 5C). The known bacterial species most closely related to this OTU were *Clostridium celatum*, *Clostridium saudiense,* and *Clostridium disporicum*, all isolated from the feces of mammals, including humans, swine and rats (*Horn, 1987*; *Angelakis et al., 2014*; *Agergaard et al., 2016*). Although in most *Clostridium* spp. studied so far, tetracycline resistance is conferred by the *tet*(M) gene (*Adams et al., 2002*), *tet*(W) has been detected in a medical isolate from *Clostridium difficile* (*Spigaglia, Barbanti & Mastrantonio, 2008*). A similarity search using this gene as a query showed the detection of homologues in various uncultured environmental *Clostridium* spp. (Supplemental Information 2). The other two most correlated OTUs (Otu00045 and Otu00062) both belonged to the genus *Ruminococcus*_1, closely related to *Ruminococcus callidus* and unclassified *Ruminococcus* spp., respectively. All of these have been reported in human and other mammal feces (*Wang, Cao & Cerniglia, 1997*; *Leser et al., 2002*; *Schmidt et al., 2011*). Another OTU of interest was OTU00008, which represented up to 1.75% of the total community (0.87% in average). OTU00008 was closely related to *Ruminococcus bromii*, one of the predominant species in human, swine, and cattle gut (*Moore, Cato & Holdeman, 1972*; *Klieve et al., 2007*). Although the genus *Ruminococcus* was originally considered as

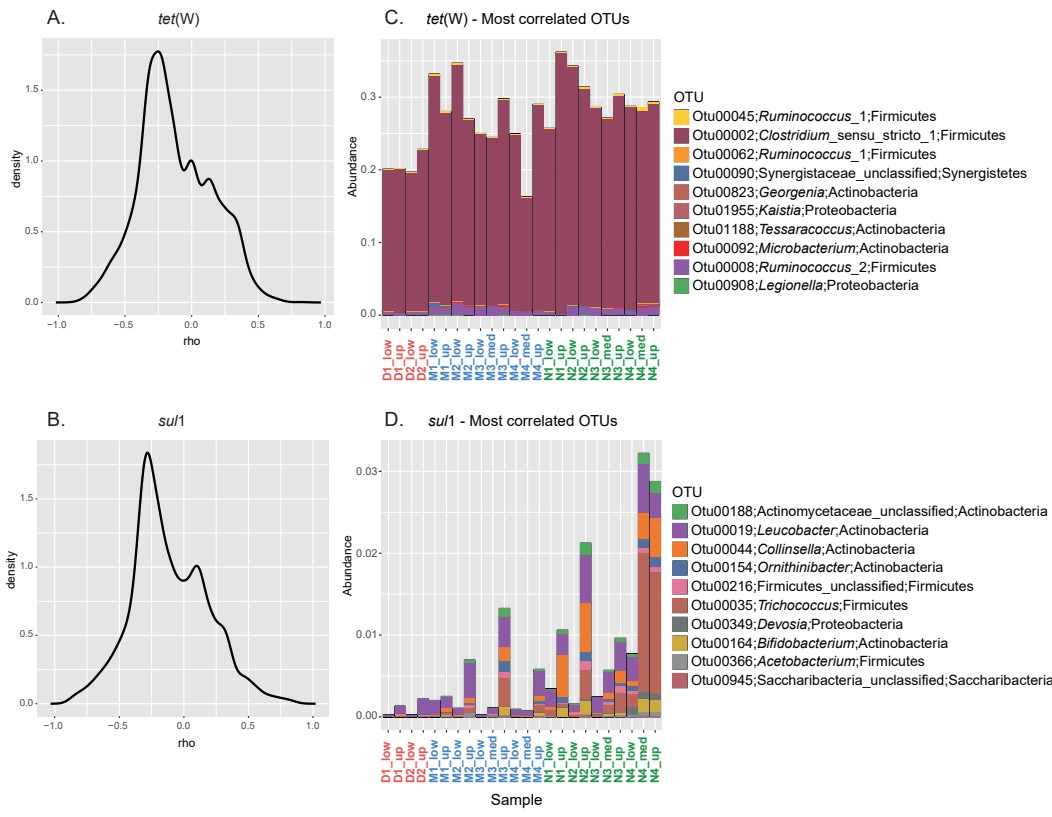

**Figure 5** **Correlation between specific OTU and frequency of the ARG *tet*(W) and *sul*1.** Kernel density curves representing the frequency of the Spearman's correlation coefficients calculated between ARG and OTU and relative abundance of the 10 most positively correlated OTUs for each ARG. (A) Kernel density curves for *tet*(W). (B) Kernel density curves for *sul*1. (C) Relative abundance of OTUs correlated with *tet*(W). (D) Relative abundance of OTUs correlated with *sul*1.

to comprise non-spore forming species, a recent report has demonstrated the production of spores by *Ruminococcus* isolated from human colon and rumen (*Mukhopadhya et al., 2018*). This gives additional support to our approach and shows the importance of defining the sporobiota based on its properties, rather than on the reports of spore production in cultured isolates.

In spite of the widespread phylogenetic distribution of *sul*1 (*Aminov, 2010*), the 10 most *sul*1-positively correlated OTUs belonged mainly to Actinobacteria (Table 2). The other most correlated OTUs belonged to Firmicutes, Proteobacteria, and Saccharibacteria (all with a correlation coefficient above 0.8). Unsurprisingly, all these highly correlated OTUs were closely related to organisms commonly reported as part of the human/mammal microbiome and/or are commonly found in WWTPs. Compared to *tet*(W), none of the best correlated OTUs represented a major fraction of the sporobiome community (<0.25% in average, Fig. 5D). However, one has to be careful in the interpretation of these correlations. The system under study is mainly driven by the influence of the WWTP. Most *sul*1-correlated OTUs were related to Group 1 and their abundance correlates with high

**Table 2  Correlations between OTU relative abundance and ARG frequency.** Each line indicates the correlation coefficient of the 10 most positively correlated OTUs, their associated *p*-values (with and without adjustment for multiple comparison) and the phylogenetic identity of OTUs. Adjusted *p*-value were calculated using the Benjamini–Hochberg method to control the false discovery rate.

| OTU | Cor. coeff. | *p*-value | Adj. *p*-value | Phylum | Genus |
|---|---|---|---|---|---|
| *tet*(W) | | | | | |
| Otu00045 | 0.841 | 2.58E−07 | 8.23E−04 | Firmicutes | *Ruminococcus_1* |
| Otu00002 | 0.805 | 2.10E−06 | 1.47E−03 | Firmicutes | *Clostridium_sensu_stricto_1* |
| Otu00062 | 0.712 | 9.42E−05 | 1.06E−02 | Firmicutes | *Ruminococcus_1* |
| Otu00090 | 0.673 | 3.09E−04 | 2.15E−02 | Synergistetes | Synergistaceae_unclassified |
| Otu00823 | 0.673 | 3.13E−04 | 2.15E−02 | Actinobacteria | *Georgenia* |
| Otu01955 | 0.667 | 3.66E−04 | 2.32E−02 | Proteobacteria | *Kaistia* |
| Otu01188 | 0.663 | 4.19E−04 | 2.50E−02 | Actinobacteria | *Tessaracoccus* |
| Otu00092 | 0.651 | 5.64E−04 | 2.91E−02 | Actinobacteria | *Microbacterium* |
| Otu00008 | 0.638 | 8.05E−04 | 3.52E−02 | Firmicutes | *Ruminococcus_2* |
| Otu00908 | 0.635 | 8.68E−04 | 3.62E−02 | Proteobacteria | *Legionella* |
| *sul*1 | | | | | |
| Otu00188 | 0.882 | 1.18E−08 | 1.83E−05 | Actinobacteria | Actinomycetaceae_unclassified |
| Otu00019 | 0.856 | 9.86E−08 | 5.66E−05 | Actinobacteria | *Leucobacter* |
| Otu00044 | 0.850 | 1.45E−07 | 5.80E−05 | Actinobacteria | *Collinsella* |
| Otu00154 | 0.843 | 2.41E−07 | 6.70E−05 | Actinobacteria | *Ornithinibacter* |
| Otu00216 | 0.840 | 2.86E−07 | 7.62E−05 | Firmicutes | Firmicutes_unclassified |
| Otu00035 | 0.835 | 3.79E−07 | 8.64E−05 | Firmicutes | *Trichococcus* |
| Otu00349 | 0.827 | 6.42E−07 | 1.17E−04 | Proteobacteria | *Devosia* |
| Otu00164 | 0.826 | 6.61E−07 | 1.17E−04 | Actinobacteria | *Bifidobacterium* |
| Otu00366 | 0.821 | 8.69E−07 | 1.42E−04 | Firmicutes | *Acetobacterium* |
| Otu00945 | 0.816 | 1.18E−06 | 1.76E−04 | Saccharibacteria | Saccharibacteria_unclassified |

concentrations of $C_{org}$ and $N_{tot}$ and other environmental indicators of the direct influence of treated wastewater release (Fig. S5).

## DISCUSSION

Our results show that the effluents from the studied WWTP have a clear impact on the deposition of ARG associated with the bacterial sporobiome in sediments. Higher concentrations of ARG were found in the proximity of the outlet pipe compared to a more distal zone. These results were expected given the reported effect of treated wastewater on the levels of ARG in WWTP-impacted environments. In a study evaluating the dissemination of ARG from a river receiving WWTP discharge and farming runoffs, a decrease of ARG abundance with distance to the source was measured, reflecting a decreasing anthropogenic impact. However, most ARG were detectable even in distant sediments (*Chen et al., 2013*). Another study investigating the spatial distribution of different ARG in relation to WWTP outflow into a lake detected an abundance up to 200 fold higher in the vicinity of the outlet pipe with exponential decline (ARG abundance and concentration) as a function of the distance to the WWTP outlet pipe (*Czekalski, Gascón Díez & Bürgmann, 2014*).

The ARG investigated in the present study appeared to be differently enriched within the sporobiota, which was reflected by their unequal distribution in the sediments. Our results also suggest that the abundance of *tet*(W) in the sporobiota community is greater than *sul*1. This contrasts with previous studies quantifying ARG abundance within the total bacterial DNA, which have shown that the abundance of *sul* 1 was higher (or at least as abundant) than for other ARG, including *tet*(W) (*Munir, Wong & Xagoraraki, 2011*; *Czekalski, Gascón Díez & Bürgmann, 2014*; *Guo et al., 2014*; *Rodriguez-Mozaz et al., 2015*). In order to generalize this interpretation, the analysis of additional ARGs within the sporobiota needs to be considered in the future. This is even more relevant when taking into account the fact that the patterns of dissemination of ARG associated with the sporobiota appears to be unique, as suggested by the abundance of *tet*(W) across the area investigated. For instance, the low $R^2$ calculated for the linear regression between ARG abundance and distance (Fig. S2) demonstrated that ARG spatial distribution associated with the sporobiome is not only a function of distance to the outlet pipe (highest frequency of *tet*(W) found in the middle area). In contrast, a previous study on total microbial community DNA reported the highest relative abundance (copy number normalized by 16S rRNA gene copy numbers) of *tet*(W) in the vicinity of the wastewater discharge and a significant logarithmic decay in relative abundance with distance (*Czekalski, Gascón Díez & Bürgmann, 2014*). This suggests that *tet*(W) accumulation is specific to the sporobiota community. Interestingly, highest *tet*(W) frequencies were found among samples that coincided with a high level of particulate metal concentration in sediments (Figs. S1B; S3). Correlation between ARG and metal concentration has been observed in various environments (*Berg et al., 2010*; *Knapp et al., 2011*; *Ji et al., 2012*), including a previous study in Vidy Bay (*Devarajan et al., 2015*). However, it should be noted that not all the samples with high metal concentration have a high frequency of *tet*(W), suggesting that there might be other reasons for the correlation observed rather than co-selection. Moreover, co-selection is not expected for the sporobiota if the former is found in a metabolically inactive state (spore), but only in the case of the vegetative growing state.

Persistence and dissemination of ARG within the sporobiota appeared to be linked to a small number of species. The abundance of these ARG-bearing microorganisms in the human/animal microbiome and subsequent selection processes will strongly influence the potential of ARG to accumulate and spread in the environment. Our results show that endospore-forming Firmicutes play an important role in the environmental dissemination of certain ARG and thus, their potentially long-term persistence in the environment. Firmicutes are expected to be a major component of the sporobiota because of their ability to form endospores (*Schleifer, 2009*). In addition, Firmicutes might constitute a proxy for anthropogenic impact as they are common in the human microbiome (*Browne et al., 2016*). Moreover, this phylum includes a wide variety of pathogens, especially within *Clostridia* (*Popoff & Bouvet, 2009*).

Our results pointed to a *Clostridium* species as a potentially important vector for *tet*(W) dissemination and accumulation. This provides experimental support for the implication of the sporobiota and the sporobiome in the spread of antibiotic resistance (*Tetz & Tetz, 2017*; *Bengtsson-Palme, Kristiansson & Larsson, 2017*). The results are in

accordance with a previous study investigating the natural reservoirs of antibiotic resistance in the human gut, which revealed that *tet*(W) was preferentially present within Clostridiaceae, Ruminococcaceae, and Lachnospiraceae (*De Vries et al., 2011*; *Van Schaik, 2015*). Interestingly, the sample with the lowest relative abundance of Firmicutes was also the one with the lowest abundance/frequency of *tet*(W), suggesting that the abundance of Firmicutes in the sporobiome was a good predictor of *tet*(W) contamination.

Compared to *tet*(W), a larger fraction of the community was correlated to *sul*1. Notably among those were OTUs affiliated to Actinobacteria (Table 2). A recent study reported an increase in Actinobacteria abundance that correlated to an increase in *sul*1 frequency in activated sludge under tetracycline and sulfamethoxazole selection pressure, suggesting *sul*1 is well represented in Actinobacteria (*Zhang et al., 2016*). In addition to Actinobacteria, OTUs belonging to Firmicutes, Proteobacteria, and Saccharibacteria were also among the most correlated to *sul*1 frequency. This can be seen as a confirmation of the distribution of *sul*1 among a broad taxonomic range of bacterial clades, which is reflected in the sporobiome. The association of sulfonamide resistance with Class 1 Integrons and its transfer through horizontal gene transfer might explain this broad taxonomic distribution (*Aminov, 2010*).

The selection of ARG may already occur within human microbiota. It is now largely accepted that human intestinal bacteria not only exchange resistance genes among themselves, but can also interact with other bacteria in the colon (*Salyers, Gupta & Wang, 2004*). This makes the human gut a potential reservoir of antibiotic resistance genes for pathogenic bacteria (*Sommer, Dantas & Church, 2009*). Resistant spore-formers are then partially released with human feces and arrive in WWTPs, where (if in a vegetative state) they have the opportunity to mix and engage in horizontal gene transfer with other bacteria. If the final treatment is not sufficient to eliminate these organisms, they end up in the environment where they can interact with environmental organisms (*Baquero, Martínez & Cantón, 2008*).

## CONCLUSION

The presence of ARG in the sporobiome highlights that it may be difficult to remove the legacy of resistance genes once released into the environment. Detecting ARG within this intrinsically long-lasting fraction of the microbial community also has important implications for the monitoring of ARG and the understanding of the processes explaining host-to-environment-to-host spread of antibiotic resistance. Notably, several studies have reported the transport and survival of thermophilic Firmicutes endospores in cold marine sediments, demonstrating their high dispersal and survival propensity (*Hubert, 2009*; *de Rezende et al., 2013*). Likewise, comparison with our previous study on a sediment core taken in the same basin but not directly influenced by the wastewater treatment discharge (Rhone catchment area; *Madueño et al., 2018*) also showed the enrichment of the sporobiota in the same resistance genes, providing further evidence that the sporobiota is a significant vector for ARG transport over considerable distance (Rhone river catchment). Moreover, the temporal analysis performed in our previous work suggests that the sporobiota

preserves the signature of ARG over long time scales (decades) (*Madueño et al., 2018*). Even if environmental conditions become harsh, this highly resilient fraction of the bacterial community can survive for long periods of time and may eventually re-enter humans via drinking water or other routes of contact, which may constitute an important long-term risk for human health.

### Funding

This work was supported by the Swiss National Science Foundation grant number CR23I2_162810. The funders had no role in study design, data collection and analysis, decision to publish, or preparation of the manuscript.

### Grant Disclosures

The following grant information was disclosed by the authors:
Swiss National Science Foundation: CR23I2_162810.

### Competing Interests

The authors declare there are no competing interests.

### Author Contributions

- Christophe Paul performed the experiments, analyzed the data, prepared figures and/or tables, authored or reviewed drafts of the paper, approved the final draft.
- Zhanna Bayrychenko and Sevasti Filippidou performed the experiments, analyzed the data, authored or reviewed drafts of the paper, approved the final draft.
- Thomas Junier performed the experiments, analyzed the data, contributed reagents/materials/analysis tools, authored or reviewed drafts of the paper, approved the final draft.
- Karin Beck performed the experiments, contributed reagents/materials/analysis tools, authored or reviewed drafts of the paper, approved the final draft.
- Matthieu Bueche contributed reagents/materials/analysis tools, authored or reviewed drafts of the paper, approved the final draft.
- Gilbert Greub analyzed the data, authored or reviewed drafts of the paper, approved the final draft.
- Helmut Buergmann conceived and designed the experiments, analyzed the data, contributed reagents/materials/analysis tools, authored or reviewed drafts of the paper, approved the final draft.
- Pilar Junier conceived and designed the experiments, analyzed the data, prepared figures and/or tables, authored or reviewed drafts of the paper, approved the final draft.

### DNA Deposition

The following information was supplied regarding the deposition of DNA sequences:
Sequencing data was deposited to NCBI under the Bioproject accession number PRJNA396277.

## Data Availability

The raw data are included as Tables or as Supplemental Files.

## Supplemental Information

Supplemental information for this article can be found online at http://dx.doi.org/10.7717/peerj.4989#supplemental-information.

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
