# Peer review of "Dissemination of antibiotic resistance genes associated with the sporobiota in sediments impacted by wastewater"

_PeerJ, doi:10.7717/peerj.4989_

## Round 0.1 · original submission · Major Revisions

Your article has been reviewed by three experts in the field. I agree with their comments, and share the concerns of one of the reviewers regarding the statistical analyses reported in your manuscript. Please make it clear in your revised manuscript why Spearman correlation was used rather than Pearson correlation. Also correct your P values for multiple testing, indicating whether you used familywise error rate or false discovery rate for correction. Provide the unadjusted and adjusted P values in Table 2 of your revised manuscript. To your boxplots, please add dot plots so the distribution of the acutal data points can be observed (this can be done with jitter and boxplots() in the R environment).

·

Basic reporting

As the study is of importance in various fields; this data would be of importance to the researchers working in similar lines. However, few more inputs are required.
1. It is necessary to correct the usage of the terms Sporobiota and Sporobiome through the text. Sporobiota - is the collection of sporeforming bacteria. Sporobiome - is the collection of genes of sporeforming bacteria.
For example: lines 53-53, states as "In terms of dissemination, the so-called
54 “sporobiome” is a portion of the bacterial community of great interest."

However it is should be: "In terms of dissemination, the so-called
54 “sporobiota” is a portion of the bacterial community of great interest."

The same issues should be resolved throughout the text.

2. Line 158-159, primer's sequeces shouldl be listed.

3. Line 213 "sporobiome DNA" - my suggestion is to take DNA off. For example no one use "microbiome DNA", as microbiome/sporobiome is DNA totality.

Experimental design

The research desing match well with the goal, results and discussion. The report is clearly written, with a logical flow. Experimental design aims to answer the research questions.

Validity of the findings

The report is clearly written, with a logical flow. Authors describe an important data showing exploring new ways of antibiotic resistance spreading due to the presence of highly resistant spores. The work is of high importance as sporobiota and sporobiome represent a particular challenge for ARG spreading

·

Basic reporting

No comment

Experimental design

The experimental design is sound, although it is a pity that they have not included more resistance genes in their analysis. However, I have several questions regarding the statistical methods (see comments below), which would need to be addressed or clarified before publication. Particularly I am worried that the correlations they report between resistance genes and taxa may not statistically hold.

Validity of the findings

The implications of the sporobiome in the dissemination of antibiotic resistance has barely been investigated before, making this an important inroad into an under-investigated topic. The study is very small in scope, covering only two antibiotic resistance genes. Overall the conclusions are supported by the data. That said, I feel that the authors could caution a little bit more on the generality of their findings, as they only investigate two antibiotic resistance genes and in some sentences suggest that what they see for those two would be valid for all or most resistance genes.

Additional comments

- It would have been interesting to see a comparison between the ARGs in the sporobiome and those in the full community. I don't think this is totally necessary for the study to be publishable, but since the authors seem to have those results already, I think a supplementary figure comparing the two would add some more value to this study at a very low time investment for the authors.

- Line 57-59: This is a very important point, and I am very happy that the authors point this out explicitly.

- Line 74-75: What would actually promote HGT in aquatic sediments? Sure it can happen, but this does not seem to be one of the most important environment for transmission of ARGs.

- Line 85: I suggest adding the word "potentially" after "thus".

- Line 102: What was the rationale for selecting those two specific genes (tetW and sul1)? There are a multitude of tet resistance genes, and sulfonamides are not that widely used in human medicine (in Europe), so I think this would need some further motivation.

- Line 111-112: I think "spatial distribution of ARG tet(W) and sul1" is strangely formulated. Please rephrase.

- Line 122: What type of cores?

- Line 145: Is this supposed to say "21.460 x g"?

- Line 146: I am not sure what "with 1 volume of 70% ethanol" means

- Line 196: Please specify which environmental parameters already here.

- Line 196-199: To me the use of Spearman correlation seems inappropriate as both those collections of numbers convey important abundance information, which I believe is important for the interpretation the authors make. Please motivate the use of Spearman correlation rather than Pearson correlation. Also, were the correlations corrected for multiple testing? There are a lot of tests being done here, so I think controlling the false discovery rate is important in this context!

- Line 199-201: What was the use of these density plots?

- Line 203 and 208: What standardization method was used?

- Line 222: Actually, the copies of ARG per g of sediment does not necessarily reflect the total ARG pool. With only two genes is very hard to estimate the total ARG abundance, and tetW and sul1 are not the most suitable genes for that either (I have data to back this statement up, but unfortunately it is still unpublished).

- Line 224-225: Note that the ARG frequency may also be biased due to differential ratios of bacteria in the samples, which should be pointed out.

- Line 232-233: Are the p values for correlations corrected for multiple testing? Otherwise, that should be done. Also, what correlation measure what used for this? Spearman?

- Line 236-237: This is unexpected to me, but interesting. My guess would have been that distance would explain a lot of the variation in ARG abundance and frequency.

- Line 238: Remove "a" before "different"

- Line 239: Change "pattern" for "patterns".

- Line 241: Remove "the" before "dilution"

- Line 246-248: Without further evidence for this, I think the tetW-metal connection is a rather far fetched speculation.

- Line 254-255: This seems like an important finding to me, that deserves more highlighting. At the same time, comparative data from the same sample in the total community composition would really be needed to draw firm conclusions (see earlier comments). Could that be included?

- Line 262: Add a comma after "community".

- Line 263: The question associated with the definition of "resilience to the extraction method" is what the degree of noise is, i.e. how many bacteria that persist the treatment for random reasons. Has that been assessed?

- Line 270-271: To say this more samples would be needed from an even more remote area of the bay to compare to, as even the distant samples look pretty much the same. I would refrain from drawing this conclusion from this data alone.

- Line 272-273: However, this could aslo be driven by similar selection conditions all across the lake. This could be an instance of species sorting driven communities OR dispersal driven. See the metacommunity work in e.g.:
Leibold MA, Holyoak M, Mouquet N et al. (2004) The metacommunity concept: a framework for multi-
scale community ecology. Ecology Letters, 7, 601-613.
Loreau,M. et al. (2001) Biodiversity and ecosystem functioning: current knowledge and future challenges. Science, 294, 804-808.
Bengtsson J. Applied (meta)community ecology: diversity and ecosystem services at the intersection of local and regional processes. In: Verhoef HA, Morin PJ (eds.). Community Ecology: Processes, Models, and Applications. Oxford: Oxford University Press, 2009, 115-30.

- Line 277-279: It could equally well be a pretty small fraction originating from the WWTP, because the variation between samples seems to be pretty small. This conclusion is overstated with the evidence presented.

- Line 281-283: With regards to Clostridia and Bacilli, I agree with the authors that the conclusion is supported.

- Line 287: Shouldn't this be the genera most associated with proximity to the WWTP rather than the most abundant? The most abundant ones were virtually the same regardless of distance to the WWTP.

- Line 290-292: I can't find this in any of the figures, where is this data shown?

- Line 293-294: The samples with high metal concentrations are also among the most nutrient rich samples. Couldn't that be an alternative/complementary explanation than metal exposure? This fits with the observation that WWTP-associated organisms are enriched in those samples as well.

- Line 314 (Table 2): Please provide p-values (corrected for multiple testing) for the correlations in table 2

- Line 323-324: Note that it is not known from these results if any of these genera are actual carriers of tetW, or if they just happen to correlate for other reasons. I would really refrain from this speculation.

- Line 338-339: Partially, confirmation of tetW presence in these genomes could be done by just searching a genome database for tetW. Therefore, if the authors really want to speculate about dissemination, such a search would really improve the support for their argument.

- Line 342-344: I actually think this is more surprising for sul1 than for tetW. Sul1 is (I think) more widespread among environmental bacteria.

- Line 350-352: This is true also for tetW, so the speculation above is... well, quite speculative.

- Line 370: Change "sporobiome community abundance tet(W) in greater amounts than sul1" into "sporobiome community abundance of tet(W) is greater than sul1".

- Line 371-373: This is the reason why it is a pity (and a big limitation) that only two resistance genes were studied!

- Line 382-383: Unclear if this decay appeared in this study or the cited one. Rephrase and clarify.

- Line 385: What is "TM"?

- Line 391-392: What is to be expected? Unclearly formulated.

- Line 393-407: I would like to know if these correlations are actually significant after correction for multiple testing (see earlier comments). Otherwise I would say that this statement is not supported by the results.

- Line 411-412: This part I agree on. What I am suspicious about is the specific species/genes correlations pointed out earlier.

- Line 413: This seems like it could be random effects. Please check the significances!

- Line 417-422: See previous comment.

- Line 435-436: This is an important point which could be stressed more.

- Line 439: Change "Comparison" to all lowercase.

- Line 439-440: "study on a sediment core taken away from the influence of a local contamination source" It is unclear what this part of the sentence means. Please rephrase.

- Line 443-446: This is a very good and important point and I would like to stress that this conclusion holds even if the correlations between taxa and resistance genes reported turn out to be non-significant.

- Figure 1: I think that this figure is somewhat non-essential and could potentially be moved to the supplement. I have no strong opinion on this, however.

- Figure 2: What is tested for in the global model? Relation between distance and abundance/frequency? Is this a linear model?

- Figure 4: It is impossible to distinguish genera with the same colors. Please use different symbols in addition to the colors to make this plot readable.

·

Basic reporting

For the most part the paper is reported well. I have suggested a number of edits as there were sentences that were less transparent than I would have liked.

The literature cited is appropriate.

Data has been made available and the supplementary material is helpful.

Experimental design

The aims and objectives are clear. The authors do make an effort to contextualise their study in the literature. There could be more clarity in the methods when it comes to the associated data about the site. There is no mention of where this data comes from.

Validity of the findings

The findings can be more clearly stated at times and I have made suggestions below where that might be possible.

Additional comments

The following are specific comments on the manuscript:
Is the 'Sporobiome' a vector for ARG dispersal (as stated in line 30) or is it something that is drawn
upon by many vectors and disseminated? I think the latter.

I think the abstract should read (line 36), "...the dissemination of tetW and sul1, two common antibiotic
resistance genes." Or simply leave out the latter bit...it's not correct the way it's currently worded
as it's two genes in total, not two tet genes and sul.

Line 37. You cant have 'spatial distribution analysis of ARG in the sporobiome DNA', you can only have it across a landscape. This section needs rewording.

It's not clear how important a 'sporobiome' could be for humans or the environment if by definition it is
not active. I imagine what the authors mean is that spores are a potentially important fraction of the
community that can have atypical dissemination characteristics which could allow for greater
dissemination of ARGs than would be predicted by metabolically active and growing cells.

Line 92. 'associated with'
Line 97-99. Surely ARGs and ARBs are everywhere in the lake. I challenge you to find a litre of water that does not contain an ARG/ARB.

Line 185, commas are more commonly found on the bottom not the top (i.e., 100,000 not 100'000)

Line 203. The parameters stated here are not mentioned earlier and so the abbreviations need to be explained.

Line 236. Clarify what is meant by 'R2-adj'

Line 258. Can you clarify what is meant by M4_med,

perhaps say what it means and then put this in ()

Lines 300-304. This needs to be reworded to clarify the point being made.

Lines 374. This sentence needs to be reworded, I'm not sure what it means.

Line 383. Can you restate this sentence--i'm not sure what is meant.

Line 385. What does 'TM' mean?

Line 391. I think you should extend your definition of where the sporobiome is metabolically inactive, as it seems that it should extend to the samples you acquired, which are not found in a wwtp.

Line 393. Is it appropriate to say that ARGs are accumulating in the sporobiome? Perhaps you can say
that there is selective loss of ARGs not in spores and a concommittant persistance of ARGs in spores.

Line 398. I don't thin the 'heat-resistance' of Firmicutes is relevant to this study.

Line 404. Should be 'pointed to'.

Line 439. lowercase 'comparison'.

---

## Round 0.2 · Minor Revisions

I agree with the comments of Johan Bengtsson-Palme. Please address his comments and submit your revised manuscript at your earliest convenience.

·

Basic reporting

No comments

Experimental design

The authors have addressed my comments satisfactorily. I am happy that the p-values seem to hold for multiple testing correction.

Validity of the findings

The conclusions are supported by the data, and in the revised manuscript the language is more well-balanced in terms of interpretations of the findings.

Additional comments

The authors have addressed almost all of my concerns very satisfactorily, either by reasonable explanations, straightening out misunderstandings or performing additional analyses. I still have two minor points that I think should be clarified.

1. I think there was a misunderstanding about my comment on the biases of ARG frequency and abundance (line 244-251 in the revised manuscript). To be clear, what I meant is that I agree with the authors that total ARG abundance could be biased by the differential microbial load in sediments and the proportion of different bacterial groups. However, what I wanted the authors to acknowledge is that ARG frequency the number of copies of ARG per ng of extracted DNA) can ALSO be biased by differential abundance of different bacterial groups. Please add this note to avoid confusing other scientists designing similar studies.

2. I don’t really but the authors’ explanation that “the fact that Tet(W) has several domains related to Elongation factor Tu and G (...) makes it difficult to verify the identity of the blast hits obtained.” First of all, the supplied BLAST searches makes it clear that they were carried out using very permissive identity cutoffs. I re-did the same searches and filtered the data for >90% sequence identity and e-value < 1e-170. This filters out most of the hits that would be ambiguous, particularly for sul1. In these searches I only found the Actinomycetaceae family from the sul1 correlations, and the Ruminococcus and Clostridium genera from the tet(W) correlations.
Given that the authors assume that their primers are specific enough to only detect sul1 and tet(W) (and not the elongation factor proteins), such stringent cutoffs would be reasonable. Note that these results indicate that using correlations between taxa and resistance genes is a flawed approach to finding carriers of resistance genes and at best provides some possible candidates. I personally do not think that there is any substantive support for the speculation about resistance gene carriers in the “Correlation tests between OTUs and ARG” paragraph (lines 329-374), and that it could be substantially shortened as it does not (again in my opinion) provide that much value to the study.

---

## Round 0.3 · accepted · Accept

Thank you for addressing the remaining queries of the reviewer. I hope you will consider PeerJ for future publications resulting from your research.

#